# Reasoning about concepts with LLMs: Inconsistencies abound

**Rosario Uceda Sosa, Karthikeyan Natesan Ramamurthy, Maria Chang &
Moninder Singh**
IBM Research
Yorktown Heights, NY 10598, USA
{rosariou,knatesa}@us.ibm.com,maria.chang@ibm.com,moninder@us.ibm.com

## Abstract

The ability to summarize and organize knowledge into abstract concepts is key to learning and reasoning. Many industrial applications rely on the consistent and systematic use of concepts, especially when dealing with decision-critical knowledge. However, we demonstrate that, when methodically questioned, large language models (LLMs) often show significant inconsistencies in their knowledge.

Computationally, the basic aspects of the conceptualization of a given domain can be represented as Is-A hierarchies in a knowledge graph (KG) or ontology, together with a few properties or axioms that enable straightforward reasoning. We show that even simple ontologies can be used to reveal conceptual inconsistencies across several LLMs. We also propose strategies that domain experts can use to evaluate and improve the coverage of key domain concepts in LLMs of various sizes. In particular, we have been able to significantly enhance the performance of these LLMs with openly available weights, using simple KG-based prompting strategies.

## 1 Introduction

Conceptualization is a key cognitive ability that enables abstract thinking. Through concepts we communicate and learn complex knowledge, generalizing from instances and applying learned principles to new situations. Conceptualization sits at the base of symbolic reasoning and allows us to plan ahead and innovate beyond our physical experience.

For example, children can easily conceptualize 'chair' to the point of identifying new instances of chairs they haven't seen before. Furthermore, when they learn 'armchair', they innately understand it is a type of chair (Is-A hierarchy) and that whatever principles we apply to 'chair' also apply to its sub-concept 'armchair'. In summary, children innately learn not only the concepts themselves, but their associated Is-A hierarchies and how to reason about them *consistently*.

Such consistent use of conceptualization is critical in several industrial applications where LLMs are used. Take, for example, a customer-facing chatbot in a property and casualty insurance company. It has to be dependable in its knowledge of, say, vehicle types (Koutsomitropoulos & Kalou, 2017): if a 'vehicle' is defined as an 'insurable object' that is covered according to a 'policy', the LLM should consistently assert that 'cruiser motorcycle', 'van' or 'scooter' are vehicles but a 'child's tricycle' is not considered a vehicle for the purpose of an insurance policy. Any inconsistency in identifying other sub-concepts (related by the IsA or subConceptOf relation) of 'vehicle' in this context could lead to a lack of trust in the system and downstream harm to users.

It is this consistent use of and reasoning about a concept hierarchy by LLMs that we want to evaluate and discuss. Provided that an LLM has already some knowledge about concepts (and sub-concepts) in a given domain, we ask, is this knowledge dependably shown when answering direct questions? Can we correct any inconsistencies with additional context? Can we leverage the LLM's knowledge *consistently* in simple reasoning tasks, for example, concluding that a 'cafe racer' and a 'naked bike' are both types of motorcycles and that all

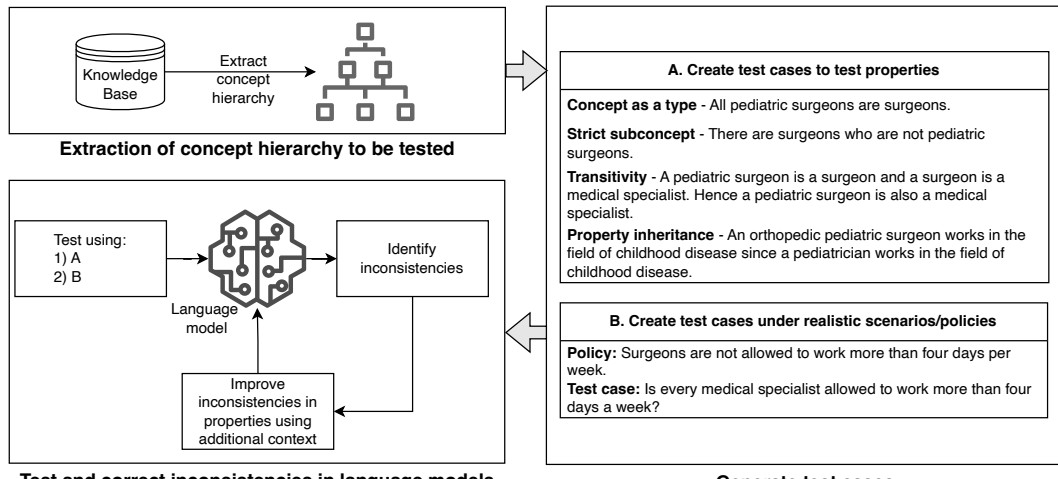

Figure 1: Our proposed approach to test and correct the inconsistencies in an LLM's knowledge of concept hierarchies and in its application to realistic scenarios.

properties of a motorcycle (like having a maximum capacity for two passengers) apply to both of them, as well as all other sub-concepts of 'motorcycle'?

We propose a systematic and automatic method to test and correct inconsistencies, as outlined in Figure 1. It consists of a three step process: (1) we automatically extract a concept hierarchy to be tested from a knowledge base; (2) we generate -also automatically- various test cases to sieve inconsistencies via direct questions (Figure 1A) that leverage reasoning about these concepts (Figure 1B) and (3) we test the language model to identify inconsistencies and reduce them using generated additional context.

We believe that the relation between KGs and LLMs is at the heart of a neuro-symbolic approach to AI. KGs provide structured, factual information in an algorithmic, traceable way, while LLMs offer advanced natural language understanding and generation. As interest on adapting LLMs to specialized domain vocabularies is growing (Zhang et al., 2023; Shen et al., 2024), the integration of these complementary technologies holds the potential for creating more accurate and reliable AI systems, specially in applications requiring both precise information and sophisticated language capabilities (AlKhamissi et al., 2022). That is why, in general, we are interested in the various aspects of the integration of both KGs and LLMs, including knowledge validation.

It is also worth noting that, at this point, it is not necessary to consider more complex relations or reasoning patterns, since we already found plenty of inconsistencies at the most basic levels of concept reasoning in all studied LLMs. Our goal is not to produce a thorough vetting of a concept graph in an LLM, but to create a systematic baseline for the fine-grained evaluation of inconsistencies. Understanding why these happen and how to fix them in practice promises to be a fertile area of research.

Our main contributions are: (1) We devise methods for using ontologies to assess the consistency and coverage of conceptualization in LLMs - this is done by creating test cases based on the knowledge graphs (KGs) or ontologies in an automated manner, (2) we demonstrate that several well-known LLMs with openly available weights demonstrate many inconsistencies in their knowledge, even with very rudimentary, small ontologies, and (3) we show that using simple prompting approaches we can reduce these inconsistencies and improve the coverage of domain concepts in several LLMs with openly available weights.

Our paper is structured as follows. We start with a working definition of conceptualization (Section 2), then show how to extract a sample ontology from Wikidata for our evaluation

(Section 3). We define the inconsistencies we look for in LLMs (Section 4), and discuss a use case where we test the consistency in reasoning performed by LLMs for this ontology (Section 5). The results of our evaluation are discussed in (Section 6), followed by related work (Section 7) and conclusions/future directions (Section 8). Additional experimental details and results for one more domain (finance) are presented in the appendix. The datasets needed to reproduce our results along with prompts that we use are included in the supplemental material.

## 2 A working definition of conceptualization for KGs

We define a concept $C$ as a set of its instances. For example, 'medical specialist' describes all the people whose professional occupation is a medical specialty. Subconcepts like 'surgeon' or 'pediatrician' represent subsets of medical specialists. A *subConceptOf* (also known as IsA) hierarchy of concepts is the simplest incarnation of an ontology, where every node represents a concept and the directed edges represent the subConceptOf relationship. This directed graph reflects a 'mental picture' of the domain that users would expect to be stable and consistent.

Here, we consider the key computational properties of conceptualization shown below:

- **Concept as a type**. If $A$ is a subConceptOf $B$ then every instance of $A$ is an instance of $B$. Paraphrasing, every $A$ is also a $B$, or an $A$ is a type of $B$. E.g., all pediatric surgeons are surgeons.
- **Strict subconcept property**. When the subConceptOf relation is strict, there are instances of $B$ that are not instances of $A$. E.g., there are surgeons who are not pediatric surgeons.
- **Transitive property**. The relation subConceptOf is transitive. *I.e.*, if $A$ is a subConceptOf $B$, and $B$ is a subConceptOf of $D$, then $A$ is a subConceptOf of $D$. E.g., Given that a pediatric surgeon is a surgeon and a surgeon is a medical specialist, a reasonable user would infer that a pediatric surgeon is also a medical specialist. In particular, when we apply the transitive property to a concept graph, we are effectively adding implicit edges to those already explicit in the graph, like the edge between a pediatric surgeon and a medical specialist. The resulting, augmented set of edges is usually called the *deductive closure* of the graph.
- **Subconcept property inheritance**. Every property that $B$'s have, $A$'s also have. E.g., if we assert that 'medical specialists must be board certified', we would also expect that surgeons and pediatric surgeons need to be board certified.

There are other properties that apply to conceptualizations (like reflexivity, or specific semantics of concept properties), but we consider that the properties above sum up the behavior that most users would expect when reasoning about concepts, as well as rules and constraints, which also require the use of subConceptOf hierarchies.

Even though this is an informal discussion about concepts and how most people would reason about them, we must remark that in the first property above we are equating the set theoretical definition of a concept (i.e., the set of its instances) with type theory (a concept is also a type). Most people won't have trouble understanding the context in which the term 'surgeon' is used, and we expect that an LLM would do likewise.

## 3 A Wikidata-based sample ontology

We start with a small ontology fragment automatically extracted from Wikidata (Vrandečić, 2012; Erxleben et al., 2014; Vrandečić & Krötzsch, 2014) from a set of seed concepts. Wikidata is a well known KG, whose content is agreed upon by thousands of contributors. We chose a common vocabulary, namely, medical specialties and specialists ('surgeon', 'pediatrician', etc.) as shown in Figure 2. For the sake of conciseness, we have not drawn all of the subConceptOf relations between the medical specialties, but they are part of the underlying deductive closure of the KG.

We tried this small vocabulary with openly available LLMs, and we found that all of them answered correctly questions about the edges in over 90% of our dataset question clusters, designed to test the edges and paths of the knowledge graph (see Section 6). This fact alone demonstrates that the LLMs 'knew' of the graph vocabulary. Also, it is worth noting that our results are not domain specific, as we have obtained similar ones in other domains. In Appendix B we show a simple example from personal finances.

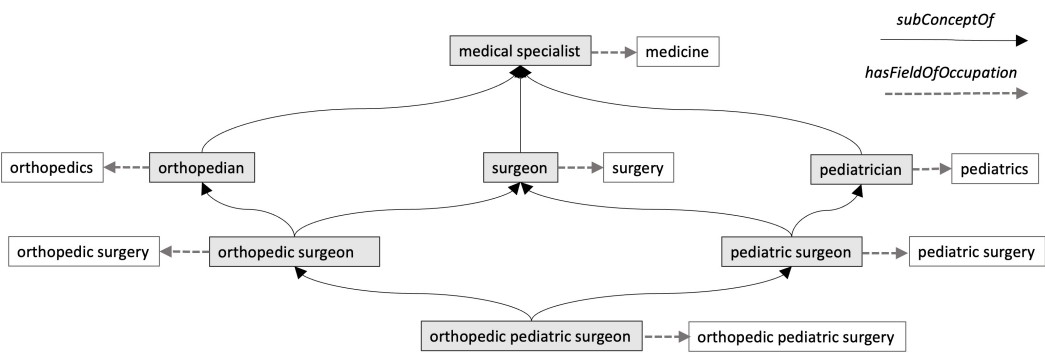

Figure 2: A concept hierarchy snapshot: medical specialists and their specialities.

To extract the ontology, we provided two seed concepts, the main entity, 'medical specialist' (Q3332438), and one sample property to test the property inheritance, 'medical specialty' (Q930752). We get a graph with 3130 nodes (medical specialists and their occupations).

From this initial graph, it is straightforward to extract the segment that includes their instances (P31) and subclasses (P279). Given that the differences between both of them are arbitrary in a higher order graph like Wikidata (where entities can be instances of a class and yet have instances themselves), we subsume them into the subConceptOf relation.

It is worth repeating that reasoning about these concepts means being able to assert not just the explicit edges, but also their deductive closure, i.e., the virtual, implied edges by the subConceptOf relation as defined above.

## 4 Logical and ontological inconsistencies and question clusters

The simplest form of logical contradiction is to both assert and deny the exact same fact, e.g., "a cardiologist is a medical specialist" and "a cardiologist is not a medical specialist". Other common form of logical inconsistency consists in asserting a fact and denying one of its (more or less immediate) consequences with respect to a given set of axioms or properties. For us, this set is described in Section 2 above. This definition aligns with the formal definition of consistency discussed in Nguyen (2008) – "in knowledge-based systems the notion consistency of knowledge is often understood as a situation in which a knowledge base does not contain contradictions."

Starting with the KG, we automatically generate a set (or cluster) of queries with a uniform yes/no expected answer. The questions in each cluster map to statements that must collectively be true or false (depending on how the cluster is designed). A set of mixed answers reveals an inconsistency with respect to the conceptualization properties above, which can be checked automatically and renders the cluster *inconsistent.*

However, there is the possibility that an LLM answers 'no' for an entire cluster for which the expected answer is 'yes'. This could be due to knowledge missing in the LLM. We call these *incomplete clusters* instead of *inconsistent clusters* (where answers are a mix of both 'yes' and 'no'). As we see in Section 6 the former are exceedingly rare (or non existent in several LLMs), meaning the LLMs tested 'know' these concepts.

While it is not possible to algorithmically sieve all the knowledge in LLMs, even using standard heuristics that exist to determine sets of unsatisfiable statements (statements

which cannot possibly be all true at the same time and thus reveal an inconsistency), our approach allows an end user to define mission critical concept hierarchies and test them to ensure consistent responses for them. These graphs are small and test the key properties of concepts, as the general problem of identifying minimal unsatisfiable sets in KBs (equivalent to inconsistent clusters) is NP-hard (McAreavey et al., 2014; Gernert, 2005; Pan & Zhang, 2007).

Before we describe in detail these question clusters, we emphasize that all LLMs tested have responded correctly to some of their individual questions in over 98% of the clusters (that is, we have very few incomplete clusters), even with a simple prompt. This means that the LLMs can process both the vocabulary as well as the linguistic forms shown here.

Each of the cluster types defined below test one or more of the conceptualization properties defined in Section 2.

**Edge clusters** We check the first two properties in Section 2 with edge clusters.

The first type of edge cluster is called a *positive edge cluster*. As the name indicates, it refers to an edge that exists in the KG. We use various expressions in order to test the LLM linguistic flexibility and robustness. Take, for example, the edge (surgeon, subconceptOf, medical specialist). The corresponding positive edge cluster consists of the following questions:

- Is a surgeon a medical specialist?
- Is a surgeon a type of medical specialist?
- Is every surgeon a medical specialist?
- Is a surgeon also a medical specialist?

Obviously, the expected answer to all of these is 'yes'.

If an LLM answers all these questions in the negative, it is possible that it hasn't been trained or doesn't know this particular edge (i.e., it's an incomplete cluster). However, if say, all questions except the third question are answered 'yes', there is obviously an inconsistency in the LLM knowledge. If every surgeon is NOT a medical specialist, it cannot be that a surgeon IS a medical specialist or that a surgeon is a type of medical specialist. That is, these answers imply an unsatisfiable set of statements. These questions are very simple and, in theory, it would be possible to increase the variations in questions in each of the clusters, but it would only likely make the model responses more inconsistent according to our definition, so we consider that the performance we report here is an optimistic estimate.

*Inverse edge clusters* are used to test the *strict subconcept property* above, when a concept A is strictly contained in its parent B, meaning that there are instances of B that are not instances of A. For example, for the inverse of the positive edge cluster above, a subset of the questions we ask would be:

- Is every medical specialist a surgeon?
- Is a medical specialist a type of surgeon?

All these questions can be generated automatically by comparing the instances (P31) of both A and B and checking there is no 'same as' property (P460) between them (which is exceedingly rare, anyway).

The third type of edge cluster, the *negative edge cluster* tests the first set theoretic property of conceptualization, but for non existent (false) edges. These 'false' edges are automatically built from the hierarchy by selecting random nodes not related by the subConceptOf relation. For example, (cardiologist, subConceptOf, dermatologist) or (surgeon, subConceptOf, hypnotherapist). The questions are linguistically formulated as in the *positive edge clusters*. In this case, a subset of the questions we have would look like:

- Is a surgeon a hypnotherapist?
- Is a surgeon a type of hypnotherapist?

It is important to note that some of the questions where the LLM disagrees with the ground truth answer in our dataset may be technically correct. An LLM may object to one of these particular linguistic forms and may make a well reasoned argument for its answer. For example, when asking "is an orthopedic pediatric surgeon an infection control physician?" the language model (mixtral-8x7b-instruct in this case), instead of a 'yes' or 'no' answer, offers an explanation for a non-committal answer: "it is possible that an orthopedic pediatric surgeon may work in the field of infection control, however this is not their primary field of occupation, which is orthopedic surgery and pediatric surgery". This answer is technically correct, but not *consistent* with the answers to the majority of similar questions such as, "is an orthopedic pediatric surgeon a infectious disease physician ?" or 'is an orthopedic pediatric surgeon a hepatologist?", and dozens of other similar questions that this LLM answered with a simple 'no'. Still, given that we are not testing the LLM knowledge, but its *consistency*, we still have to mark this answer -when compared to the majority of similar answers- inconsistent.

**Path clusters**   This second type of cluster tests the transitivity of the subConceptOf relation described in section 2 by querying a sequence of edges in a given path. Using the same linguistic forms as before, we ask about the deductive closure of a path (the curved arrows in fig 3). In our sample graph there are 4 such paths. Two of these are, *[orthopedics pediatric surgeon, pediatric surgeon, surgeon, medical specialist]*, as shown, and *[orthopedics pediatric surgeon, orthopedics surgeon, orthopedian, medical specialist]*. The edges in the paths are queried as shown in the case of the edge clusters.

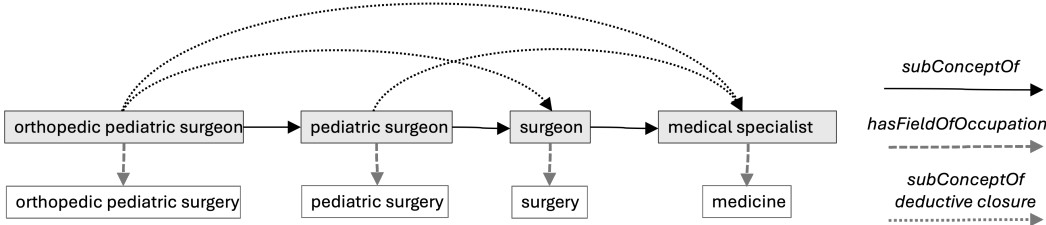

Figure 3: Deductive closure between orthopedic pediatric surgeon and medical specialist.

**Property hierarchy clusters**   The last type of cluster tests the fourth property of conceptualization, *subconcept property inheritance*. This is a core feature of conceptualization that affords abstract reasoning. For example, consider the questions below:

- is the field of occupation of a surgeon surgery?
- is an orthopedic pediatric surgeon a surgeon ?
- is the field of occupation of a orthopedic pediatric surgeon surgery?

If the field of occupation of a surgeon is surgery, and an orthopedic pediatric surgeon is a surgeon, we would expect that the field of occupation of an orthopedic pediatric surgeon is also surgery. Of course, a more specific answer is that orthopedic pediatric surgery is the occupation of an orthopedic pediatric surgeon, but the fact remains that all of the models tested answer the above cluster in the affirmative in the majority of cases. Again, it is the matter of consistency that concerns this study. Again, the edges in this subgraph are tested using linguistic forms as in edge clusters.

## 5   Demonstrative use case

Why is it important to ensure that an LLM can consistently answer seemingly simple questions about the edges of a given KG? Imagine a set of policies, rules or processes that a health care network or an insurance company wants to define and use in an AI application. Take, for example:

1. "Only pediatric surgeons can perform surgery on patients younger than 18 years old."

2. "Only surgeons are required to work no more than four days per week."

Not only the policy designers would expect to define and manage these rules using abstract concepts, but the users of the application would expect to query these policies using more specific vocabulary related to their case. Consequently, the application should be able to *understand* whether a pediatric surgeon or a pediatrician satisfy either policy.

We have created a small dataset of 10 scenarios with simply worded policies that apply to the medical specialists in our sample knowledge graph (included in our supplemental materials). Each scenario is tested with two types of questions. The first one is "Does the policy apply to every {specialist}?" where {specialist} is substituted by one of the 7 terms in our sample graph ('pediatrician', 'surgeon', 'orthopedic surgeon' and so on). The second type of question mimics the policies above, using the same type of term substitution. The queries corresponding to the policies above are:

1. Is every {specialist} allowed to treat or operate on patients younger than 18 years old?

2. Is every {specialist} allowed to work more than four days per week?

Knowledge of our sample graph, or the equivalent implicit knowledge, is required to answer these straightforward questions correctly, which shows that concept hierarchies lie at the base of this type of industrial applications. However, as we see in Table 1 many of the LLMs with openly available weights get many individual answers wrong, even though they also get some answers right. It is worth noting that there are no 'incomplete' scenarios (where every individual question in a cluster is incorrect) here. So, we ask ourselves again, what happened? Is it lack of specific knowledge (one particular edge or node) or lack of overall consistency in the knowledge? Why do the LLMs fail to answer correctly in some cases and not in others? Can we pinpoint the specific *holes* in the knowledge so they can be corrected?

To dig deeper into these questions, we need to generate a dataset to test systematically the knowledge graph directly, as we have discussed in Section 4.

Table 1: Evaluation of 10 policy-based scenarios (14 questions per scenario).

| LLM name | % incorrect individual answers (140) | % inconsistent scenarios (10) |
|---|---|---|
| google/flan-t5-xl (Chung et al., 2022) | 65.71 | 100 |
| google/flan-t5-xxl (Chung et al., 2022) | 24.28 | 90 |
| google/flan-ul2 (Tay et al., 2023) | 15 | 70 |
| meta-llama/llama-2-13b-chat (Touvron et al., 2023) | 22.8 | 80 |
| meta-llama/llama-2-70b-chat (Touvron et al., 2023) | 15 | 60 |
| tiiuae/falcon-180b (Almazrouei et al., 2023) | 15 | 60 |
| mistralai/mistral-7b-instruct-v0-2 (Jiang et al., 2023) | 13.57 | 60 |
| mistralai/mixtral-8x7b-instruct-v0-1 (Mistral.AI, 2023a) | 13.57 | 40 |
| thebloke/mixtral-8x7b-v0-1-gptq (Mistral.AI, 2023b) | 35 | 100 |

## 6 Evaluation and coverage improvement

The three types of clusters described above are designed to highlight the inconsistencies of the LLM knowledge. We automatically extract them from the topology of the test KG above, producing 119 clusters, with 96 edge clusters (the high number is due to the fact that we have negative and inverse edge clusters representing edges NOT in the graph). More details are provided in Appendix A, and additional results for a different domain ontology are presented in Appendix B.

We test this graph in 9 openly available models (see Table 1 for model information) using a simple prompt with 11 sample questions from the medical domain. These models are hosted in our own organization's infrastructure. The prompt used is provided in the supplementary materials. We ask for yes/no answers which can be automatically tallied. A 'yes' answer means that, for every possible instance, the question can always be answered in the affirmative. Otherwise, the answer should be 'no', as it doesn't hold for the concept (i.e., all its instances). The results are displayed in Table 2. For conciseness' sake, we have added all the edge clusters together. A few facts worth noting. First, we notice in the leftmost column that there are very few incomplete edges (where all the individual responses in a cluster are wrong). This means that out of the 96 edge clusters, the vast majority of them are *known* to the LLMs. In fact, some LLMs have no incomplete edges. Second, we notice that the notion of property inheritance is the most challenging, since all of the models fail over 36% of the time.

Table 2: Evaluation results by model using a simple prompting strategy.

| LLM name | % incomp. edges (96) | % incons. edges (96) | % incons. paths (12) | % incons. property inherit. (11) | % all incons. (119) |
|---|---|---|---|---|---|
| google/flan-t5-xl | 4.17 | 40.62 | 16.66 | 36.36 | 41.18 |
| google/flan-t5-xxl | 1.04 | 35.41 | 16.66 | 36.36 | 34.45 |
| google/flan-ul2 | 4.17 | 26.04 | 33.33 | 54.54 | 32.77 |
| meta-llama/llama-2-13b-chat | 0 | 13.54 | 16.66 | 36.36 | 15.97 |
| meta-llama/llama-2-70b-chat | 3.13 | 22.91 | 16.66 | 45.45 | 26.89 |
| tiiuae/falcon-180b | 0 | 17.7 | 16.66 | 36.36 | 19.33 |
| mistralai/mistral-7b-instruct-v0-2 | 0 | 4.16 | 25 | 36.36 | 9.24 |
| mistralai/mixtral-8x7b-instruct-v0-1 | 2.08 | 22.91 | 16.6 | 36.36 | 25.21 |
| thebloke/mixtral-8x7b-v0-1-gptq | 1.04 | 32.29 | 16.66 | 36.36 | 31.93 |

Table 3: Evaluation results by model with context-augmented prompts

| LLM name | % incomp. edges (96) | % incons. edges (96) | % incons. paths (12) | % incons. property inherit. (11) | % all incons. (119) | % improve. (all incons.) |
|---|---|---|---|---|---|---|
| flan-t5-xl | 1.04 | 10.41 | 25 | 27.27 | 14.29 | **26.89** |
| flan-t5-xxl | 1.04 | 10.41 | 0 | 0 | 9.24 | **25.21** |
| flan-ul2 | 1.04 | 12.5 | 0 | 27.27 | 13.45 | **19.33** |
| llama-2-13b-chat | 0 | 7.29 | 0 | 9.09 | 6.72 | **9.24** |
| llama-2-70b-chat | 2.08 | 10.41 | 0 | 9.09 | 10.92 | **15.97** |
| falcon-180b | 1.04 | 13.54 | 0 | 0 | 11.76 | **7.56** |
| mistral-7b-instruct-v0-2 | 0 | 6.25 | 0 | 0 | 5.04 | **4.20** |
| mixtral-8x7b-instruct-v0-1 | 0 | 9.37 | 0 | 27.27 | 10.08 | **15.13** |
| mixtral-8x7b-v0-1-gptq | 0 | 9.375 | 0 | 27.27 | 10.08 | **21.85** |

Next, we look to enhance the performance of the initial prompt by adding to the queries a context with the propositionalization of the knowledge that was missed by all the models, i.e., we use the same context for all the models. This context is computed automatically, as our underlying dataset (included in supplemental file) maps the cluster questions into their corresponding assertions. For example, 'is every orthopedic surgeon a surgeon?' is mapped to 'every orthopedic surgeon is a surgeon'. This allows us to generate the context for queries on a second test. This 'wholesale' approach to context augmentation yields roughly the same improvements as if we tailored the context to each individual model.

With this simple prompt augmentation strategy, we obtain a sizable performance enhancement as shown in Table 3. The rightmost column reflects this performance enhancement in the clusters, showing that now points of inconsistency have been reduced up to one third. It is worth noting that even leveraging this explicit knowledge doesn't eliminate inconsistency altogether.

# 7 Related Work

Seminal work by Petroni et al. (2019) demonstrated that a language model could learn relational knowledge (i.e. facts one would expect to be found in a knowledge base) during pre-training. This raised the possibility that language models could serve as approximations for knowledge bases right out of the box. However, Elazar et al. (2021) used paraphrased querying to show that such knowledge could not elicited consistently/reliably. This led to the development of frameworks for measuring inconsistency in language models (Jang et al., 2021; Laban et al., 2023; Sahu et al., 2022) as well as novel training setups with consistency-based loss (Elazar et al., 2021). The consistency issues found in LLMs have been identified as one of the key areas of future work needed to enhance LLMs so they share the same strengths -and consistency- as KBs (AlKhamissi et al., 2022).

Large language models have recently been shown to exhibit abilities akin to 'reasoning' when prompted in certain ways. For example, chain-of-thought prompting ( Wei et al. (2022)) gets models to provide explicit steps it took to arrive at an answer. Nevertheless, it is not clear whether it actually demonstrates that the LLMs are actually reasoning Wei et al. (2022); Kojima et al. (2023). Wang et al. (2023) explores the consistency of LLM results via chain-of-thought and studies ways of making such results more consistent. A nice survey on the current state of knowledge in reasoning in LLMs is provided by Huang & Chang (2023). Other work has looked what LLMs actually know Yin et al. (2023); Srivastava et al. (2023); Sun et al. (2023) and have shown that LLMs exhibit are very weak in this regard, with performance sometimes barely surpassing random guessing Srivastava et al. (2023).

Improving consistency and factual correctness of language models is related to ongoing work that aims to integrate external knowledge into LLMs, either from unstructured sources like retrieved documents or from structured knowledge bases (Feng et al., 2023; Yang et al., 2024). Approaches may be applied at different stages of the model lifecycle (Pan et al., 2024): KGs may be used in pre-training (Yasunaga et al., 2022), tuning (Zhang et al., 2024; Cheng et al., 2023) or information from KGs can be incorporated directly into the prompt (Andrus et al., 2022; Fatemi et al., 2024).

Our proposed approach differs from the above related works in that we perform analysis of consistency of knowledge of LLMs with respect to a small and targeted KG by automatically generating test cases in the shape of query clusters. Our clusters can act as building blocks of satisfiability -or unsatisfiability-, so we can identify small portions of knowledge to edit or evaluate. Our generated query clusters measure the knowledge consistency of simple edges, both positive (e.g. 'true') and negative (e.g. 'false'), as well as the consistency of paths and property inheritance reasoning, in contrast to other approaches which do not discriminate the semantics of relations or take into account the deductive closure of the graph (Rajan et al., 2024).

Also, we do not require an externally annotated dataset, such as a QA benchmark. We also perform targeted editing of the LLM's knowledge using prompting. This is because in industrial applications, the domain expert requires consistency in a relatively small fragment of a specialized KG. For example, in a general KG, a bicycle is objectively a type of vehicle, but in our introductory insurance example, bicycles are typically not covered by vehicle insurance and so they cannot be considered vehicles per the insurance contract. This means that domain experts may need to edit the knowledge. While KG reasoning and editing in general may be useful like in GraphRAG [1] or (Luo et al., 2024), we explore more targeted editing that can be systematically tested and verified to gain the trust of the domain experts and other relevant stakeholders.

---

[1]https://www.microsoft.com/en-us/research/blog/graphrag-unlocking-llm-discovery-on-narrative-private-data/

# 8 Conclusions and future work

Consistent conceptualization, especially when addressing mission critical data, is key in industrial applications. We have shown that inconsistencies creep in LLMs even when using common vocabulary and even after prompting the system with targeted content.

There are some natural future directions that emerge from these insights. The first looks to identify fine-grained knowledge issues and systematically evaluate an LLM for them. This may be done by mapping the knowledge from a KG to richer, linguistically more challenging queries that users may realistically pose to the LLM or using train-of-thought factoring of the user query into simpler queries, like the ones we produce, may help in this mapping.

The second line or research is to allow for questions that require non-committal answers and thereby handle ambiguous contexts. For example, the question 'does a pediatric surgeon always work with children?' may have a correct 'maybe' answer, as pediatric surgeons also work with teenagers. Part of establishing trust in the LLM is to ensure that ambiguous queries are properly, and consistently, dealt with. We are currently working to address these complex scenarios in various domains, like Software, Natural Disasters, Music Genres, Academic Disciplines, Occupations, etc.

## Acknowledgements

The authors thank Kush Varshney for his advice and support.

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

## A  Cluster dataset construction

Starting with the seed concept 'orthopedic pediatric surgeon', we automatically generate a data set that comprises 109 clusters, with a total of 584 questions, which include 4 different linguistic forms per query, so we have approximately 146 *semantically different* queries (some of the property clusters have 2 questions only per medical specialty). The size of the dataset is as follows:

- 15 positive edge clusters.
- 66 negative edge clusters. The number of these can be adjusted with a parameter. Obviously, in a small hierarchy, looking for unrelated nodes becomes harder if the top number is higher.
- 15 inverse edge clusters.
- 12 path clusters.
- 11 property inheritance clusters.

Each cluster, regardless of type is made up of the following:

- Expected answer. 'yes' or 'no'.
- Source. This is the source node in the directed graph
- Target. The target node of an edge or a path cluster.
- Questions. These are generated from fixed linguistic patterns for subConceptOf and for property edges. For example: "is a orthopedic pediatric surgeon a medical specialist ?", "is a orthopedic pediatric surgeon a type of medical specialist ?", "is every orthopedic pediatric surgeon a medical specialist ?" and "is a orthopedic pediatric surgeon also a medical specialist ?" for an edge cluster with source 'orthopedic pediatric surgeon'.
- Statements. The corresponding statements to the questions above: "a orthopedic pediatric surgeon is a medical specialist", "a orthopedic pediatric surgeon is a type of medical specialist", "every orthopedic pediatric surgeon is a medical specialist" and "a orthopedic pediatric surgeon is also a medical specialist". These statements are used to create augmented context to improve the consistency of the LLMs.

Our full json dataset is provided in the supplementary file.

## B  The finance domain

To prove how pervasive the inconsistencies in LLMs are, we tried several domains: government agencies, corporate occupations and finance. In this later case, we created a simple graph with just one path, so we can test the first three properties of conceptualization.

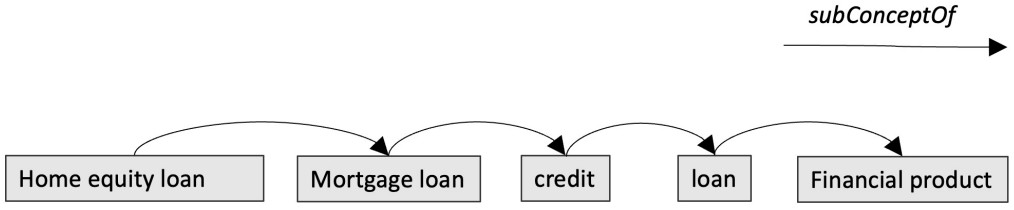

Figure 4: Finance domain: home equity loan path

This dataset has 75 edges and only 4 hierarchy clusters. The results below may not be statistically significant but we include them here because even in this Hello World example,

we get inconsistencies in similar percentages as above. Also, it is interesting that even after specifically adding the context in the prompt, we don't necessarily improve the performance in all models with respect to this simple path.

Finally, the fact that after prompting some models degrade slightly in performance (probably without statistical significance), indicates that only prompting may not be the only answer.

| LLM name | % incomp. clusters (84) | % incons. clusters (84) | % incomp. clusters (84) | % incons clusters (84) | %improve (all incons.) |
|---|---|---|---|---|---|
| google/flan-t5-xl | 2.38 | 25 | 0 | 9.33 | 15.47 |
| google/flan-t5-xxl | 4.76 | 29.76 | 0 | 12 | 17.85 |
| google/flan-ul2 | 1.19 | 25 | 0 | 10.66 | 15.47 |
| meta-llama/llama-2-13b-chat | 1.19 | 5.95 | 2.38 | 9.33 | -2.38 |
| meta-llama/llama-2-70b-chat | 1.19 | 5.95 | 2.38 | 10.66 | -3.57 |
| tiiuae/falcon-180b | 0 | 10.71 | 2.38 | 8 | -3.57 |
| mistralai/mistral-7b-instruct-v0-2 | 1.19 | 1.19 | 1.19 | 6.66 | -5.95 |
| mistralai/mixtral-8x7b-instruct-v0-1 | 1.19 | 4.76 | 0 | 1.33 | -2.38 |
| thebloke/mixtral-8x7b-v0-1-gptq | 1.19 | 11.9 | 1.19 | 0 | 10.71 |
| | Simple prompt | | Prompt augmented by context | | |

Figure 5: Finance domain: eval with simple prompt and with context

## C   Ethics statement

Our datasets were created by ourselves using publicly available wikidata ontologies. The content of our knowledge graphs is common knowledge and we do not involve any human subjects for data generation or validation.

One of the key motivations of our proposed approach is to enable users calibrate trust in LLMs and improve the consistency of LLMs in specific domains to make them more trustworthy. We believe that exhaustive testing using methods such as ours is necessary in any high-stakes application. A potential issue in using exhaustive testing methods such as what we propose is that a lot of inference calls need to be made to LLM and this increases their power consumption. However, this is mitigated by the fact that this needs to be done only for the domains of application in which the LLM is used. This testing will also reduce downstream harms for users that may happen due to inconsistent knowledge in the models.

## D   Reproducibility statement

We provide the dataset that we generated in the supplementary material. This can be used to test any model in the domains that we presented in the paper. We also discuss the methodology by which we created the dataset in sufficient detail in the paper. Any knowledgeable reader can use a similar methodology to test their own model in a domain of interest.

