# OpenReview forum: "Reasoning about concepts with LLMs: Inconsistencies abound"
_colmweb.org/COLM/2024/Conference — COLM_

### Official Review · Reviewer_Gt8x · 2024-05-07

**Rating:** 8
**Confidence:** 3
**Ethics Flag:** 1

**Summary:**

This paper seeks to address a well-known issue in LLMs: inconsistency in knowledge. The strategy is to use ontologies to extract Is-A relations and then study them in LLMs, with a focus on applicability to real-world tasks. They find that models have a lot of knowledge, but that they are inconsistent in this knowledge. They show improvements by using KB-based prompting.

This paper addresses an important topic, is clearly written, and has a promising and interesting approach. I would be happy to see it published with some revisions, but I have a few comments and suggestions.

First, a strength is that the description of ontologies (transitivity, etc) is very clear and accessible. This would I think be valuable for the audience. But, perhaps as a trade-off, I found Section 6 very rushed, with most of the important details about the key prompting method in the Appendix. I would recommend putting more detail in the main text, perhaps with an illustrative figure for the prompting method. Some of the relevant detail is in Fig 1, but it would be good to focus on exactly the prompting part.

Second, I was somewhat surprised by the discussion of not counting cases where the LLM gives mealymouthed answers like “it is possible that an orthopedic pediatric surgeon may work in the field of infection control, however this is not their primary field of occupation, which is orthopedic surgery and pediatric surgery”. This is increasingly a strategy imposed on LLMs. It’s perhaps fair (although I’m not sure) to count these as inconsistent vis a vis yes/no. But, if a lot of the apparent inconsistency comes from such examples, it would be good to know. The scenario where an LLM hedges like that in one case and simply says “no” in another feels very distinct from a yes/no asymmetry. I would address this explicitly in the main text.

A ubiquitous question: I was left wondering what, if anything, should we take away from the differences in model performance. It’s nice to test a lot of models, but are there hypotheses as to if/why some models might be better than others on this particular task.

**Questions To Authors:**

I take it cafe racer and naked bike are real examples? I wasn’t familiar with them and so was unsure if they were supposed to be “novel concepts”.

I was briefly thrown by the term “field of occupation”, which doesn’t seem idiomatic to me.

**Reasons To Accept:**

- This tackles an important topic in a clear way: making LLMs more trustworthy and consistent in the way that KBs are while retaining strengths of LMs.
- It’s clearly written and might help introduce some of the ontology-based thinking to the LLM audience in a helpful way. I found the literature reviews useful for understanding the place in the literature.
- The method seems to work and show meaningful improvements, and it seems scalable.

**Reasons To Reject:**

- Not really a reason to reject, but I’d recommend more detail on the prompting method.

- The counting of hedged LLM answers as inconsistent makes me wonder if the inconsistency is overstated.

---

> ### Author Rebuttal · Authors · 2024-05-30
>
> Thank you for your thorough review and for raising some thought-provoking questions regarding the ‘hedged’ LLMs. We're looking forward to addressing these points as we refine our paper.
>
> >More detail on the prompting method.
>
> Given that we will be given an extra page to elaborate, we will provide samples of our prompts and discuss several strategies we have used in prompting.
>
> >The counting of hedged LLM answers as inconsistent makes me wonder if the inconsistency is overstated.
>
> We have been curious about the hedged answers as well. The issue is that the LLMs were not consistent about them either. When asked with slightly different language, or referring to different edges in the graph the LLM was categoric in their answers. Out of 4600 questions that were supposed to be answered ‘no’ (across all models), these nuanced answers happened only 17 times. Even if we only count the 2700 questions about negative inverse edges (“is every surgeon a pediatric surgeon?", “is ‘surgeon’ a subtype of ‘pediatric surgeon’?”), where the system would be reasonably expected to say ‘that’s not always the case’, LLMs (all of them) didn’t answer consistently in all incarnations of the question. Hence, we considered the 17 ‘hedged answers’ as outliers for the purposes of this paper.
>
> This is not to say that a larger, more complex dataset would reveal some insight on these questions. And we may want to provide ‘automatic’ answers in our dataset (‘not always’, ‘not necessarily’ and so on) so we can take them into account. We’ll look into this as part of our future work.
>
>  >Are cafe racer and naked bike real examples?
>
> All the concepts mentioned in the paper are ‘real’, in the sense that are described (and curated by several writers) in Wikidata.
>
> CAFE RACER (https://www.wikidata.org/wiki/Q1025508)
>
> NAKED MOTORCYCLE, (synonym of) NAKED BIKE ( https://www.wikidata.org/wiki/Q1964027)

---

> > ### Comment · Reviewer_Gt8x · 2024-06-01
> >
> > Thanks for this response: I think the plan to include more details and to address hedges more sounds promising. I think addressing the usefulness and consistency of hedging here could be an additional contribution.

---

### Official Review · Reviewer_AJmS · 2024-05-10

**Rating:** 6
**Confidence:** 4
**Ethics Flag:** 1

**Summary:**

This paper devised methods that utilize ontologies to assess the consistency and coverage of conceptualization in Large Language Models (LLMs). By automatically generating test cases from knowledge graphs or ontologies, they effectively evaluate the models. Moreover, they have found that many prominent LLMs with publicly available weights exhibit inconsistencies in their knowledge, even when confronted with fundamental, small ontologies. However, through the application of straightforward prompting strategies, they have shown that it is possible to reduce these inconsistencies and improve the coverage of domain concepts across various LLMs with publicly accessible weights.

Strengths:

1.A more detailed analysis of the hierarchical confusion across multiple large language models has been conducted;
2.The paper is well-crafted, featuring lucid clarity and a straightforward logical flow that is effortless to follow.
Weaknesses:
1.The author focuses solely on hierarchical confusion while lacking necessary discussions on more comprehensive issues such as mixing of domains, inaccurate Generalizations, and contradictory information.
2.In terms of details, the analysis of the results in Table 2 and Table 3 is relatively rough. It just lists the results of different large language models on the issue without carefully analyzing the possible causes and influencing factors.

**Questions To Authors:**

Is there a connection between the inconsistency of conceptual hierarchy and the parameter size of large language models? In other words, the larger the model parameter size, the lower the possibility of inconsistency in the conceptual hierarchy will be significantly reduced?

**Reasons To Accept:**

1.A more detailed analysis of the hierarchical confusion across multiple large language models has been conducted;
2.The paper is well-crafted, featuring lucid clarity and a straightforward logical flow that is effortless to follow.

**Reasons To Reject:**

1.The author focuses solely on hierarchical confusion while lacking necessary discussions on more comprehensive issues such as mixing of domains, inaccurate Generalizations, and contradictory information.
2.In terms of details, the analysis of the results in Table 2 and Table 3 is relatively rough. It just lists the results of different large language models on the issue without carefully analyzing the possible causes and influencing factors.

---

> ### Author Rebuttal · Authors · 2024-05-30
>
> Many thanks for your helpful feedback. Your input is highly appreciated and will guide the rewrites of our paper.
>
> > Lacking a more comprehensive discussion of complex, realistic domains as well as inaccurate generalization and contradictions:
>
> We wanted to create a simple, foundational -and automatic- baseline that we can build upon in future work. While we agree there is more work ahead of us to address the nuances of incomplete and inconsistent knowledge in more complex, realistic domains, we didn’t want to obscure the glaring issues we have already found when dealing with simple concepts and basic conceptualization axioms (in section 2). Our initial goal was to test many different LLMs of various sizes on a common, simple set of concepts and a single relation and to show that, even in this limited case, all LLMs failed to deliver consistency one way or another, even though they were obviously able to process most of the semantic entailments of a concept (with respect to our axioms).
>
> >Careful analysis of possible causes and influencing factors
>
> it is clear more work is needed to come up with general enough answers. As we point out in the next response, we did look into a possible correlation between the model size (number of parameters) and inconsistencies, and while there was a negative correlation, it wasn't strong. Lines of future research include (1) Using larger datasets in several domains to give us an accurate picture of which features of an LLM affect consistency.  (2) An analysis of where the inconsistencies happen (how high in the KG hierarchy) and the interrelation between the different axioms. However,  overanalyzing a small graph, may lead to faulty conclusions. We can add these points in our discussion and future work.
>
> > Is there a connection between the inconsistency of conceptual hierarchy and the parameter size of large language models?
>
> We looked into this. There was a negative correlation between model size and volume of inconsistencies (correlation -0.36, p-value 0.046) across 32 models. However, a larger analysis across more domains would be needed to support a general conclusion. We are looking into other factors, like the topology of the network and layers or even the data they are being trained on. There is much future work to be done in this area and we believe our work sets a solid foundation for this.

---

> ### Comment · Reviewer_AJmS · 2024-06-05
> **Rebuttal**
>
> Thank you for your answers. I have no other questions.

---

### Official Review · Reviewer_71gk · 2024-05-12

**Rating:** 6
**Confidence:** 4
**Ethics Flag:** 1

**Summary:**

This paper studies the capability of LLM to reason with conceptual knowledge. It aims at testing LLM performance by evaluating conceptual consistency and the ability of handling logical inconsistencies. It proposes enhancing LLMs through knowledge integration.

**Questions To Authors:**

Would it be possible to consider reducing inconsistency and improving the coverage of domain concepts as NLI tasks and fine-tune an LLM accordingly?

**Reasons To Accept:**

- The paper is well written and illustrated by figures.
- The experiments and the use case show the potential of the proposed approach.

**Reasons To Reject:**

- Missing formal definition of inconsistency.
- Missing comparison with approaches of ontology completion (instead of adding plausible rules, removing plausible inconsistency)

---

> ### Author Rebuttal · Authors · 2024-05-30
>
> Many thanks for your review and helpful feedback. Your input is greatly appreciated and will enhance the quality of our paper.
>
> > Definition of inconsistency
>
> We define an inconsistency as a set of statements which violates one or more axioms which, in our case, are defined in section 2. For example, ‘a pediatric surgeon is a surgeon’, ‘a surgeon is a medical specialist’ and ‘a pediatric surgeon is NOT a medical specialist’ is an inconsistent set of statements with respect to the transitive property.
>
> We have designed our question clusters so that they can automatically reveal an inconsistency with respect to a given axiom when we have both expected and unexpected answers.
>
> We will amend the paper to give specific examples of inconsistencies with respect to each of the axioms in our section 2.
>
> > Comparison to ontology completion
>
> In our view, ontology completion methods are intended to solve a slightly different problem than the one we explore. While ontology completion methods seek to remedy gaps in knowledge from an ontology, our method seeks to discover gaps and inconsistencies in the latent knowledge of LLMs. Specifically, we use the ontology to systematically generate questions about concepts and properties that can be asked of an LLM in different ways. When the LLM responses to these questions are inconsistent, it indicates that the underlying LLM knowledge is inconsistent. If we have misunderstood your point, we can hopefully better understand it during the discussion period.
>
> > Would it be possible to consider reducing inconsistency and improving the coverage of domain concepts as NLI tasks and fine-tune an LLM accordingly?
>
>  Yes, it should be possible and, in fact, our datasets map the cluster questions to their corresponding knowledge statements (i.e. premises). In this paper, we wanted to define the foundational -and pragmatic- axioms for conceptualization so we could generate these datasets automatically. However, more work is needed to define the right set of entailments when knowledge is missing or inconsistent. NLI is a very interesting task to consider in this regard, although care must be taken to avoid the same inconsistent behavior from LLMs that we observe in our experiments (since state of the art NLI systems are also built upon LLMs).

---

> > ### Comment · Reviewer_71gk · 2024-06-04
> >
> > Thank you for the answers.

---

### Official Review · Reviewer_WH6c · 2024-05-13

**Rating:** 7
**Confidence:** 4
**Ethics Flag:** 1

**Summary:**

This paper presents an approach that leveraging ontologies and knowledge graphs to evaluate the conceptual knowledge understanding capabilities in several publicly available LLMs. The paper also presents simple prompting strategies to improve the domain concept knowledge in LLMs being evaluated.

**Reasons To Accept:**

This paper presents an interesting approach to assess the consistency and coverage of conceptualization in LLMs by automatically creating test cases based leveraging ontologies and knowledge graphs. Using these automated tests, the authors are demonstrate that several publicly available LLMs show inconsistent domain knowledge. The authors apply simple prompting approaches which allow the LLMs to respond with better understanding of certain medical domain concepts.

The paper is well written and easy to understand. The experimental design and evaluation setting is decently thorough.

**Reasons To Reject:**

The paper presents an approach to evaluate several LLMs (lack of) understanding of medical domain concepts using simple IS-A hierarchical knowledge. However, domain understanding requires us to deal other more complex relationships including part-of, cause/effect, inference, entailment, etc. This paper would be better accepted by the conference audience if such evaluations were presented in the paper.

---

> ### Author Rebuttal · Authors · 2024-05-30
>
> Thanks for taking the time to review our paper. We count on this insightful feedback in order to improve our work.
>
> >Dealing with more complex relationships for domain understanding:
>
> We completely agree that domain understanding requires us to deal with many more complex relationships. In fact, when we began exploring this topic, we asked models questions about more complex relations, such as those used to describe events, processes, and other kinds of dynamic knowledge. However, for the purposes of this paper, we decided to start with concepts and relations that were simple and universal enough that there could be no doubt that most LLMs would have been exposed to them during training. Additionally, we wanted to establish a foundational set of axioms that can help us in the automatic generation of datasets. Had we chosen more complex/rich/specialized/esoteric domains, we may have hidden the issues already obvious with these very basic axioms for conceptualization.  It turns out that even with these very simple graphs and vocabulary, inconsistencies are already glaring.
>
> As part of our future research, we will be working to generalize our approach to many common domains used in industrial ontologies with more, richer relations and we are looking at how to interpret and flag the inconsistencies in these more complex, interrelated concepts. We can state this rationale more explicitly in the final version of the paper.

---

### Decision · Program_Chairs · 2024-07-10

**Decision:**

Accept

**Comment:**

This paper shows an automatic way to use ontologies to create test cases that test the ability of LLMs to understand concepts. The authors report many conceptual inconsistencies and propose prompting methods for reducing their effect. All reviewers agree the paper is written clearly, has a clear message that is demonstrated convincingly and should be accepted.